# Frequency Response of RC Propellers to Streamwise Gusts in Forward Flight

**Jielong Cai ***  **and Sidaard Gunasekaran**

Mechanical and Aerospace Engineering, University of Dayton, Dayton, OH 45469, USA;
gunasekarans1@udayton.edu
* Correspondence: caij03@udayton.edu

**Abstract:** The RC propeller performance under steady and sinusoidally time-varying freestream (stream-wise or longitudinal gust) was investigated in the University of Dayton Low-Speed Wind Tunnel (UD-LSWT) in the open-jet configuration. The propellers were tested at varying incidence angles and reduced frequencies. The streamwise gust was created by actuating the shuttering system located at the test section exit and was characterized using hot-wire anemometry. A system identification model was developed for the shuttering system to determine the shutter actuation profile that would result in a sinusoidal gust in the test section. Changes in propeller thrust, power, and pitching moment were observed with an increase in propeller incidence angle under the steady freestream. The propeller's steady freestream performance was then used to predict response under periodic streamwise gusts in edgewise flight. Below a reduced frequency of 0.2, the propeller response agrees with the prediction model, suggesting that the propeller response is quasi-steady. At reduced frequencies higher than 0.2, a reduction in mean thrust and pitching moment and significant phase lag was observed.

**Keywords:** RC propeller; unsteady propeller response; streamwise gusts; edgewise flight; reduced frequency

## 1. Introduction and Background

With the usage of fixed-pitch-propeller in modern air-taxi, and urban air mobility designs [1,2], the number of investigations into propeller performance in edgewise flight has increased exponentially in recent years. While flying in low altitudes, these vehicles experience continuous or discrete changes in incidence angles between the rotor(s) and the freestream due to highly unsteady atmospheric boundary layers or wake from buildings. Several key vehicle operations such as the transition from edgewise flight to forward flight and vice-versa can take place in these unsteady conditions that can potentially lead to a catastrophic failure without sophisticated control systems. Therefore, characterizing the unsteady response of fixed-pitch propellers is crucial in developing control systems for effective gust mitigation in Unmanned Aerial Vehicles (UAVs).

### 1.1. Gust Types and Unsteady Wind Tunnels

In general, the gusts can be classified into three types: transverse, vortical, and streamwise gusts. The different types of gusts have differing effects on the force and moment response of the test article. An overview of the three different types of gusts is discussed in Jones et al. [3]. Gusts can be simulated in the wind tunnel by either moving the model in a stagnant fluid or moving the flow around a fixed model. For example, to simulate a streamwise gust, the test model is fixed in the wind tunnel test section and the fluid is accelerated around it through various actuation techniques. The streamwise gust can also be simulated by surging a fixed model at different acceleration profiles in a towing tank. The study conducted by Granlund et al. [4] compared the unsteady response of a

pitching wing using these two different methods and good agreement was found between the two methods after correcting for added mass, horizontal buoyancy, and inertial effects.

In the current study, the propeller is fixed in position and the fluid is accelerated around the propeller. The streamwise gusts in wind tunnels are usually created by actuating a series of louvers to increase the overall pressure loss in the wind tunnel circuit leading to a time-varying freestream velocity [5,6]. For a blow-down wind tunnel configuration, the louvers can be installed either upstream [7,8] or downstream of the test section [5]. For an Eiffel-type wind tunnel configuration and a closed-return wind tunnel configuration, the louvers are usually located downstream of the test section [4,6]. Due to the non-linear dependency between pressure and velocity, achieving a sinusoidally varying freestream in the test section depends on several factors such as louver location in the tunnel circuit, louver distance from the test section, and acceleration limits of the louver system [5,6,8]. Hence, as a part of this study, the shuttering system in the UD-LSWT is characterized to ensure sinusoidal variation in a streamwise gust in the open-jet test section.

Most of the aforementioned unsteady wind tunnels have a closed jet test section, except for the facility at the University of Colorado Boulder [7,8] where the testing section can be swapped between an open jet and closed jet configuration, very similar to the UD-LSWT. The characterization results indicated that the temporal variability in freestream velocity between closed-jet and open-jet configurations was significantly different. The absence of jet expansion in the closed-jet test section allowed for temporal variability in freestream that is independent of the spatial location in the test section. However, the presence of jet expansion in the open-jet test section made the unsteady freestream to be spatially dependent with increased phase lag between the control and the actuation. Similar results are expected in the UD-LSWT characterization results as well.

### 1.2. Propeller Performance in Unsteady Flows

In steady freestream conditions, the rotor blades experience highly unsteady effects such as spanwise and streamwise variations in blade sectional velocity magnitude and angle of attack that continuously change from advancing to retreating side. The majority of literature on the unsteady aerodynamics of helicopter rotors focuses on blade pitching, plunging, and fluttering due to rotor blade wake interactions [9–11] and other issues related to rotor blade dynamics. Gust encounters on an airfoil, which is a rudimentary representation of a single rotor blade have been studied theoretically [12] and experimentally [7,13] for decades. As long as the boundary layer is attached (at lower angles of attack), an increase in reduced frequency ($k$) results in a decrease in the net lift coefficient magnitude and an increase in phase lag between kinematics and lift generation. However, at higher reduced frequencies and at higher angles of attack, the formation of a leading-edge vortex is shown to cause a significant increase in lift coefficient [7,13].

In the current study, the unsteady experiments and modeling efforts were done on the entire propeller disk at different incident angles rather than focusing on the individual blades. Such investigations in the literature on the propeller response to unsteady initial conditions are extremely sparse, especially the experimental investigations. However, on a system level, a delay in the response of the drone and rotor system as well as changes in power consumption was observed [14–16] when rotors or propellers experience unsteady freestream. The performance of the multi-rotor drone is also affected by unsteady effects from the propeller-to-propeller wake interaction, and wake interaction due to the drone structure, especially in high-speed forward flight [16]. McCrink et al. [17] studied the quad-rotor performance when encountering ramp-up and ramp-down side wind gusts by conducting flight tests. A change in drone thrust up to 50 percent is measured when encountering a side gust similar to the quadrotor forward flight speed The presence of streamwise gust adds an additional layer of complexity to the already existing localized unsteady effects that have been studied extensively on helicopter rotors. The extent to which these unsteady conditions affect the propeller performance is currently unknown and is the main impetus behind the current work. The definition of the reduced frequencies

for the propeller unsteady response is not clearly defined in the literature. The current study also aims to propose a calculation method for the propeller reduced frequency that can be used to compare with the traditional unsteady aerodynamic study on flat plates and airfoils.

One of the main parameters that affect static thrust and power from a propeller is the incidence angle with respect to the freestream. Simmons and Hatke [18] investigated a three-bladed foldable propeller with a 16-inch diameter and 8-inch pitch under a steady freestream at an incidence angle, $\theta$, between 0 and 180 degrees. As compared with the classical propeller theory, the propeller thrust coefficient ($C_{Tz}$) and torque coefficient ($C_{Qz}$) decrease with the increment in propeller advance ratio ($J$) at small incidence angles ($\theta$). However, $C_{Tz}$ and $C_{Qz}$ vs. $J$ curve plateaus at higher incidence angles. $C_{Tz}$ and $C_{Qz}$ are nearly independent of $J$ at $\theta > 60°$, and shows an increase with $J$ upon approaching true edgewise flight ($\theta = 90°$). Similar results were also seen on a variable pitch propeller from McLemore and Cannon [19]. Given the strong dependability of the static propeller performance with respect to incidence angle, the unsteady response of the propeller is also hypothesized to show a similar dependency. Therefore, the present study also investigates the unsteady response of a propeller at various incidence angles.

One of the primary objectives of the current work is to determine the extent to which the propeller performance in an unsteady freestream can be predicted by a quasi-steady model. Since existing theoretical models such as blade element momentum theory and actuator disk theory tend to lose accuracy at higher incidence angles [18,19], even under steady freestream conditions, the fixed-pitch propeller response under sinusoidal time-varying freestream would introduce additional complexity. On the other hand, very few studies have conducted an experimental study on the response of fixed-pitch propellers under streamwise gust. Therefore, the current study would provide insight into the propeller gust response from a controlled experimental perspective, and determine the boundary where the propeller response becomes unsteady. Moreover, the propeller performance in a steady freestream under different $\theta$ and $J$ were used to develop a quasi-steady model that will be used to predict the propeller's unsteady response that will greatly benefit developing control algorithms for gust mitigation.

## 2. Experimental Setup

The experiments were conducted at the University of Dayton Low-Speed Wind Tunnel (UD-LSWT) lab. The wind tunnel is configured in its open jet test-section configuration. The inlet contraction ratio of the wind tunnel is 16:1, with a testing section inlet size of 0.762 m × 0.762 m. The collector downstream of the testing section is 1.370 m × 1.370 m. The steady freestream range is from 3 m/s to 37 m/s, with a turbulence intensity of less than 0.1% calibrated by a hotwire anemometry. The schematic of the experimental setup is shown in Figure 1.

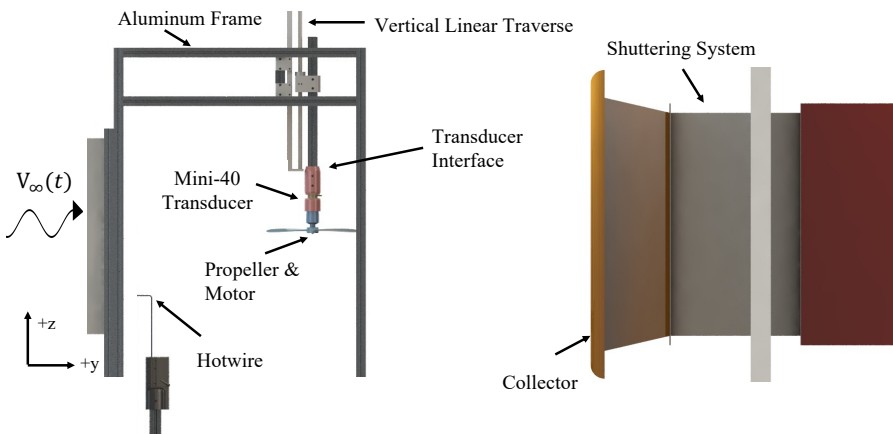

**Figure 1.** Schematic of University of Dayton Low Speed Wind Tunnel.

### 2.1. Propeller Test Setup

The propeller and motor assembly are secured at the centerline of the test section. The assembly is attached to an ATI Industrial Automation Mini-40 six-component force/torque transducer [20]. The assembly is The force balance bolted to an aluminum-extrusion interface, which was capable of changing the propeller incidence angle. The interface is mounted to a vertical linear traverse to maintain the propeller's vertical location at the centerline of the testing section when changing the propeller incidence angle, 0.610 m from the inlet of the test section. Thrust is the axial force along the z-axis of the sensor, which has a resolution of 0.02 N and an instrumentation uncertainty of 1.25%, and the normal force is along the y-axis, which has a resolution of 0.01 N and an instrumentation uncertainty of 1.0%. The propeller pitching moment is the torque measured on the x-axis, and the propeller rolling moment is the torque measured y-axis. The aerodynamic power is measured in the experiment to represent the power consumption of the propeller, which is measured as the z-axis torque. The resolution of moment measurement on all axis is 0.00025 Nm with an instrumentation uncertainty of 1.0%. The F/T balance has a maximum of 40 N calibrated range on its x and y-axis and 120 N on its z-axis. The maximum torque range is 2 Nm for all axes. The encountered thrust and torque range was 5 N to 25 N and 0.1 Nm to 0.35 Nm, respectively.

The data was sampled at the rate of 1000 Hz for all cases. A 15-s of data sampling duration is used for the steady freestream condition, while the data sapling duration changes based on each shuttering system frequency for the unsteady freestream condition. Two trials were run in each case to ensure data repeatability. To reduce the effect of sensor drift, tare values were taken before and after each test run. The experimental data would be rejected if the difference between the before and after tare value was greater than 0.1 N of force on any axis. The experiments were run again. A Finite Impulse Response (FIR) band-stop filter [21] was applied to the data to filter out the vibration frequency caused by the natural frequency of the aluminum test frame and the propeller rotational frequency. The tap tests on the test stand were performed with the propeller powered off which resulted in the structure's natural frequency of 65 Hz. The average uncertainty in the force measurement after filtration is ±9.4% across all advanced ratios tested, while the average uncertainty in torque is ±7.2%. The results presented show excellent repeatability from both experimental runs.

A Dantec Dynamics [22] P5516 Probe CTA hotwire was used to measure the instantaneous free-stream velocity experience by the propeller. To avoid interference and the induced velocity from the propeller, the hotwire was mounted more than 2 propeller diameters upstream of the propeller. The hotwire was connected to a Dantec Dynamics 54T30 Mini-CTA, and the data was collected by a National Instruments USB-6259 DAQ [23]. The hotwire was calibrated with a freestream velocity between 0–20 m/s, measured by a pitot tube and a TSI 5825 Micro anemometer. The propeller rotational speed is measured by a Thor Labs 5 mW, 532 nm wavelength laser, and a DET10A photodiode [24]. The data was also collected by the NI-USB-6259 DAQ. The sampling rate for both hotwire and the photodiode was 12,500 Hz.

### 2.2. Propeller Geometry

Two off-the-shelf, two-blade propellers are chosen for the experiment based on their experimental performance in our previous study [25,26]. The KDE 12.5 × 4.3 propeller represents a low pitch-to-diameter ratio ($\gamma/D = 0.34$) propeller which is widely used on quadcopters designed for hovering missions, while the APC 11 × 7E propeller represents a medium-high $\gamma/D = 0.63$ propeller which is used on high-speed racing drones. The performance of the APC 11 × 7E propeller under streamwise gust is also studied in our previous study at lower $K_\omega$ [26]. All propellers were driven by an E-Flite Power 60–400 Kv brushless outrunner electric motor, controlled by a Hobbywing Platinum V4 ESC under governor mode to maintain constant rotation speed. The propeller and motor were driven by a PSW 30–108 constant-voltage power supply.

The propeller local pitch ($\phi$) distribution and the local chord ($c(r)$) distribution for both propellers are shown in Figure 2. The propeller specifications are listed in Table 1. The local chord length is normalized by the propeller radius ($R$) for better comparison. The geometry data for the APC 11 $\times$ 7E propeller is provided by the manufacturer [27] directly, while the geometry for the KDE 12.5 $\times$ 4.3 propeller is measured manually. The APC 11 $\times$ 7E propeller has a higher $\phi$ when compared to the KDE 12.5 $\times$ 4.3 propeller at the same $r/R$ due to its higher $\gamma/D$ ratio. On the other hand, the KDE 12.5 $\times$ 4.3 propeller has a higher $c(r)$ closer to the root and the tip of the propeller blade than the APC 11 $\times$ 7E propeller.

**Table 1.** Propeller Geometry.

| Propeller Type | Diameter (m) | Pitch (m) | Blade Twist Angle at 75% Radius ($°$) |
|---|---|---|---|
| KDE 12.5 $\times$ 4.3 | 0.3175 | 0.1100 | 8.3 |
| APC 11 $\times$ 7E | 0.2794 | 0.1778 | 15.1 |

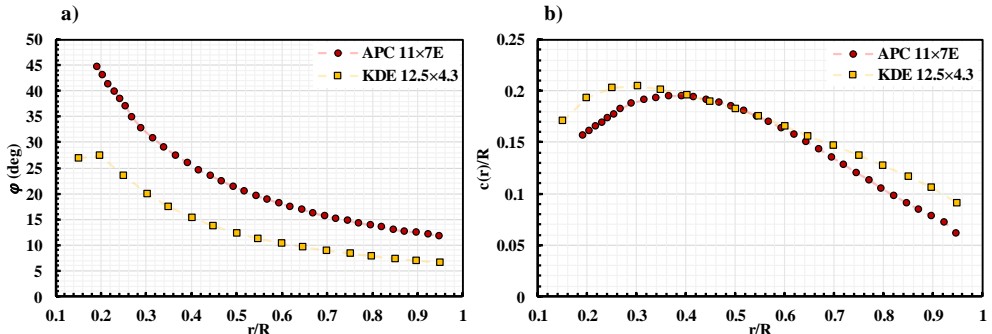

**Figure 2.** (**a**) Local blade pitch distribution and (**b**) local chord distribution for APC 11 $\times$ 7 and KDE 12.5 $\times$ 4.3 Propellers.

### 2.3. Shuttering System

Downstream of the test section, the Aerotech shuttering system is installed between the collector of the test section and the wind tunnel diffuser. A set of primary counter-rotating flat-plate vanes were mounted in a plane normal to the flow. The primary louvers operate over a louver angle ($\beta$) range of 41$°$ from full open flow to closed flow. A pair of secondary side louver vane sets were mounted in the sidewalls of the diffuser, behind the primary louvers to allow additional inflow to the diffuser and prevent wind tunnel fan blade stall. An example of the primary louvers fully open and closed is shown in Figure 3. The NI PCI-7340 motion control card [27] is used to control the position of the primary and secondary louvers. The control profile is sent by LabVIEW at a system frequency of 125 Hz, while the encoder position is also collected by the LabVIEW system at the same frequency. The shuttering system linear actuator encoder position, ATI F/T transducer, hotwire, and photodiode signal are all collected by LabVIEW on the same timestamp to ensure data synchronization.

Table 2 shows the testing matrix of the experiment. Two propeller rotational speeds were tested to investigate the Reynolds number effect and the repeatability of the experiment. To achieve different advance ratios, the propeller rotational speed is kept constant, while the wind tunnel freestream velocity is changed accordingly, between 0 and 22 m/s for the steady condition. For the time-varying freestream condition, where the shuttering system is operating, the wind tunnel fan rotational speed is kept constant. The changes in freestream velocity are purely due to the actuation of the shuttering system louver. The propeller is mounted at a 90-degree incidence angle ($\theta$), representing the low-speed hovering condition, and a 75-degree incidence angle ($\theta$), representing a nose-down forward flight condition, for the current investigation.

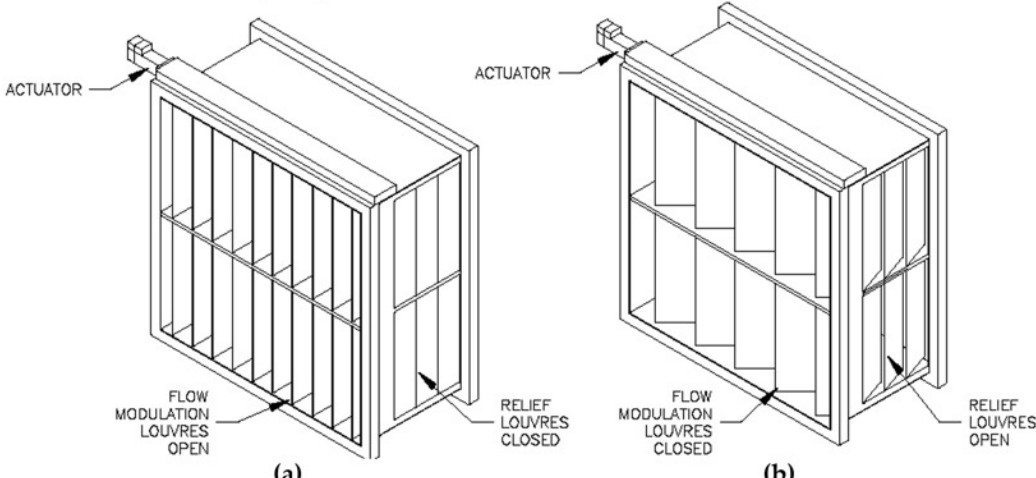

**Figure 3.** Schematic of Aerotech Suttering System in (**a**) open and (**b**) closed configurations.

**Table 2.** Test Matrix for Propeller Tests.

| Test Conditions | Values |
| --- | --- |
| Propeller RPS ($n$) | 80, 100 RPS |
| Freestream Velocity ($V_\infty$) | 0, 22 m/s |
| Propeller Incidence Angle ($\theta$) | 75, 90 Degree |
| Shuttering System Frequency ($\omega$) | 0–2.0 Hz |
| Propeller Reduced Frequency ($k_p$) | 0–0.45 |

For the time-varying freestream condition, the shuttering system is operated at a frequency ($\omega$) between 0.1 and 2.0 Hz under wind tunnel fan speeds of 200 and 300 RPM. The reduced frequency of the propeller, ($k_p$), is calculated by considering the propeller disk with the characteristic diameter $D$. This yields,

$$k_p = \frac{D}{\overline{V_\infty}}\omega, \tag{1}$$

where $\overline{V_\infty}$ is the mean velocity of the oscillating freestream.

### 3. Shuttering System Characterization

The preliminary characterization efforts of the UD-LSWT shuttering system in its open jet configuration are discussed in [28]. However, the preliminary characterization was conducted by harmonically oscillating the lover position and quantifying the changes in freestream velocity in the test section. Due to the non-linear relationship between pressure and velocity, the resultant velocity variation in the test section documented in [28] is not purely sinusoidal. To compare the test results to theoretical predictions, it is imperative to generate a sinusoidal or purely harmonic oscillation of the freestream. As such, in the current study, system identification, and closed-loop control were employed to obtain a pure sinusoidal freestream oscillation in the test section.

The Matlab system identification toolbox was used to model the wind tunnel response with an input of louver angle ($\beta$) variation, and an output of the measured freestream velocity in the test section. Several models were used to model the shuttering system and the Nonlinear-ARX model [29] has the best model fit. The model extends the traditional ARX model and includes a linear and a non-linear function to better fit the non-linear systems.

Matlab Simulink was then used to simulate a closed-loop PID control system of the shuttering system. The overall schematic of the program is shown in Figure 4. The PID controller is tuned with a relatively short response time, resulting in a high gain/acceleration. The target profile for the control system is a sinusoidal freestream velocity with the same

shuttering system frequency. The simulated louver positions were then used in the physical system in open-loop control.

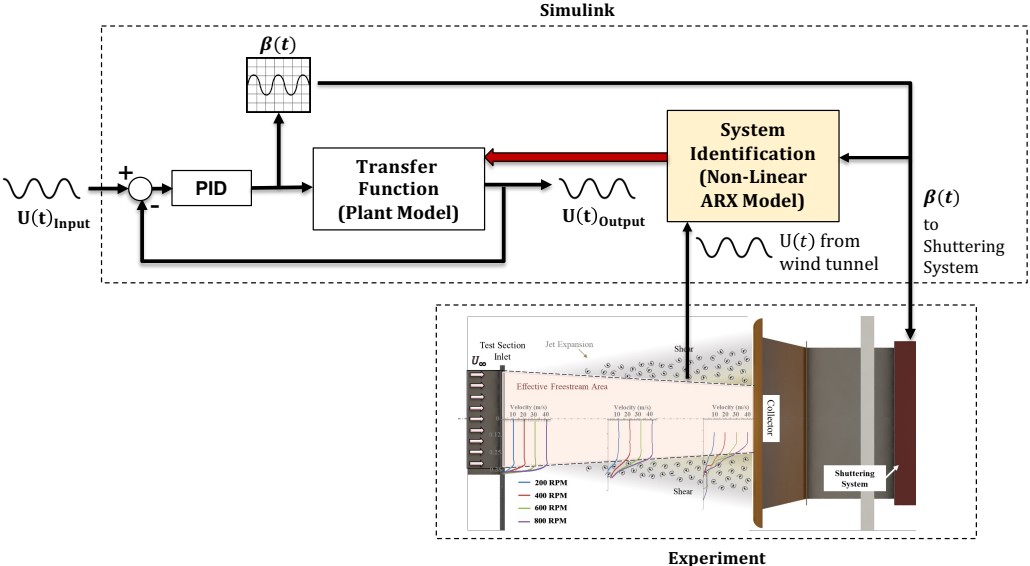

**Figure 4.** Schematic of the Matlab Simulink closed-loop PID control program.

The preliminary characterization results documented in [28] showed that the steady-state response of the freestream as a function of louver angle showed a linear relationship from $20° < \beta < 41.3°$. Figure 5 shows the normalized louver input profile and the corresponding measured wind tunnel freestream velocity. For the louver input profile, '0' represents the louver fully open (at $0°$), and '1' represents the louver fully closed (at $41.3°$). The shuttering system is operating at 0.1Hz louver operational frequency under a regular/wider louver rotation angle ($10° < \beta < 41.3°$) and limited/narrower louver rotation angle ($20° < \beta < 41.3°$). A sinusoidal model represented by $y = f(sin(kt))$ is then compared with the measured freestream velocity. The average $R^2$ value when compared to the best fit $y = f(sin(kt))$ model is 0.967 for $10° < \beta < 41.3°$ case and 0.996 for $20° < \beta < 41.3°$ case. Hence, the freestream matches the sinusoidal model better when limiting the louver operation angle. Similar trends were observed for up to a shuttering system frequency of 2.5 Hz.

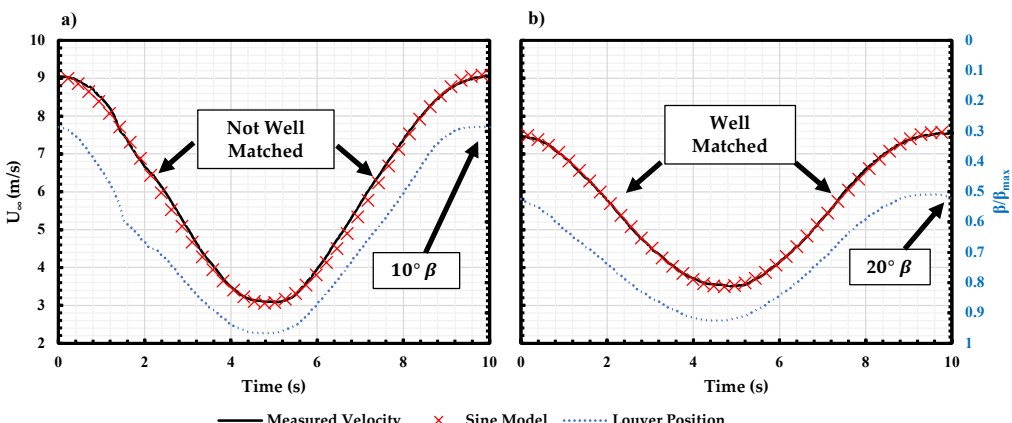

**Figure 5.** Measured freestream velocity vs sinusoidal model at 0.1 Hz louver frequency for (**a**) full and (**b**) limited louver rotation angle.

Since the system is inherently non-linear, the freestream response at each shuttering system actuation frequency is different. Therefore, the system ID process is conducted

for each louver frequency during the characterization process. Using the similar method described above, the louver rotation angle and louver position profile for each shuttering system frequency were determined to produce sinusoidal variation in freestream. Figure 6 shows the louver input and the resulting freestream velocity compared with the sinusoidal model at 1.75 Hz and 2.25 Hz louver frequencies. Note that the louver profile is non-sinusoidal for a shutter frequency of 2.25 Hz.

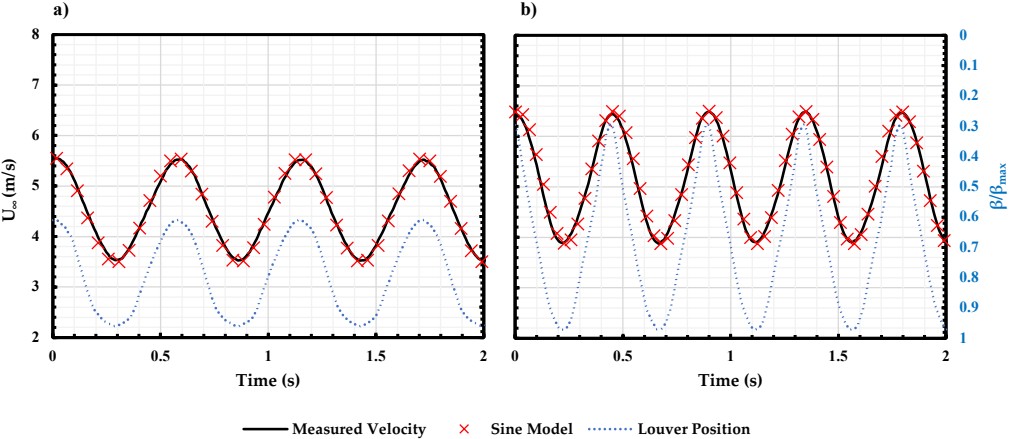

**Figure 6.** Measured freestream velocity compared with the sinusoidal model for (**a**) 1.75 Hz louver frequency and (**b**) 2.25 Hz louver frequency.

Figure 7 shows the mean freestream velocity, and the corresponding upper and lower bounds as a function of different shuttering system frequencies at different tunnel RPMs. The mean velocity of the oscillating freestream decreases with an increase in shuttering system frequency ($\omega$) between $0 < \omega < 1$ Hz. The mean velocity and the upper and lower bounds of the oscillating freestream are relatively constant at higher $\omega$. As the tunnel RPM increases, the mean velocity and the peak velocities of oscillation increase as well.

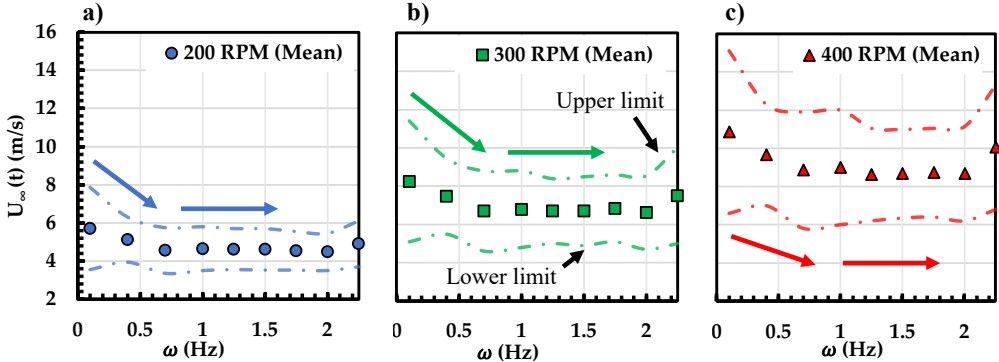

**Figure 7.** Mean, the lower and the upper bound of freestream vs. shuttering system frequency at fan RPM of (**a**) 200, (**b**) 300 and (**c**) 400.

## 4. Results

### 4.1. Propeller Performance in Steady Freestream

The propeller performance under a steady freestream will be discussed in this section for both the APC and KDE propellers at different $\theta$ as a function of $J$. All coefficients calculated in the current study followed the conventional fixed-pitch propeller methods. The steady-state results will also be used to create a quasi-steady model which can be used to compare with the results under an unsteady freestream. The steady-state of performance of the APC $11 \times 7$ propeller axial thrust coefficient, $C_{Tz}$ and power coefficient, $C_P$, between $0° < \theta < 90°$ is shown in Figure 8. Results agrees with [18,19] where a decrement in $C_{Tz}$ is observed with an increase in $J$ for $\theta < 60°$, while an increment in $C_{Tz}$ is observed

with the increment of $J$ for $\theta > 60°$. $C_P$ remains independent of $\theta$ until $J = 0.4$. At $J > 0.4$, the propeller wake angle (same as the incidence angle at low $J$) is redirected in the freestream direction due to dominant freestream velocity. This causes the $C_P$ to increase with $J$, especially at $\theta > 60°$. At lower $\theta$ cases, the $C_P$ decreases at $J > 0.4$. At $\theta = 60°$, both $C_{Tz}$ and $C_P$ remains roughly constant with $J$ which is also observed in [18].

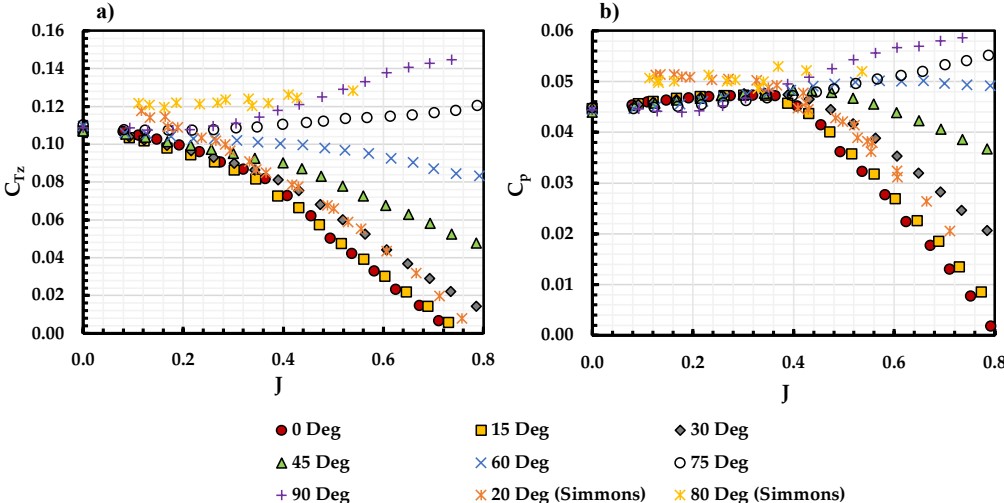

**Figure 8.** (**a**) $C_{Tz}$ and (**b**) $C_P$ vs $J$ for APC 11 × 7 under different $\theta$ at 100 RPS along with experimental results from Simmons [18].

Since the KDE propeller is particularly designed for edgewise flight, only the incidence angles between $45° < \theta < 90°$ were investigated. The experimental results for the KDE 12.5 × 4.3 propeller are shown in Figure 9. The overall trend in $C_{Tz}$ and $C_P$ resembles the result for APC 11 × 7 propeller in Figure 8. The overall magnitude of $C_{Tz}$ and $C_P$ is lower for the KDE 12.5 × 4.3 when compared to the APC propeller. This is due to a lower pitch-to-diameter ratio ($\gamma/D$) for the 12.5 × 4.3 propeller indicating a lower effective blade pitch as seen in Table 1. Moreover, $C_P$ diverges at a lower $J$ of 0.3 when compared to the 11 × 7 propeller where the divergence is seen at $J = 0.4$. This is because of the comparatively lower $\gamma/D$ of the KDE propeller when compared to the APC propeller where the thrust generated and the corresponding wake strength is lower for the KDE propeller. It is also worth noting that despite the $C_{Tz}$ for $\theta = 90°$ is higher than the $\theta = 75°$ case at $J > 0.2$, $C_P$ for the $\theta = 90°$ case is lower than the $\theta = 75°$, as the KDE propeller is designed and optimized for the edgewise flight conditions, which has a better performance at $\theta \approx 90°$. A universal trend can be obtained by using a normalized advance ratio defined as,

$$J_{z0} = \frac{V cos(\theta)}{nD} \tag{2}$$

Equation (2) considers the freestream velocity in the propeller axial direction and ignores the freestream velocity component in the propeller normal direction. Figure 10 replots the results in vs. $J_{z0}$ for the APC 11 × 7 propeller. Results for both coefficients collapse at $\theta < 60°$. At higher $\theta$, the curves do not collapse as well since $J_{z0}$ approaches 0 as $\theta$ approaches 90°. The results for KDE propellers show similar trends as well. Hence, when considering the propeller performance at $\theta < 60°$, the propeller thrust generation and power consumption can be estimated by the propeller axial freestream velocity alone. The propeller efficiency, $\eta$, can also be normalized using $J_{z0}$ as,

$$\eta_z = \frac{C_{Tz} J_{z0}}{C_P} \tag{3}$$

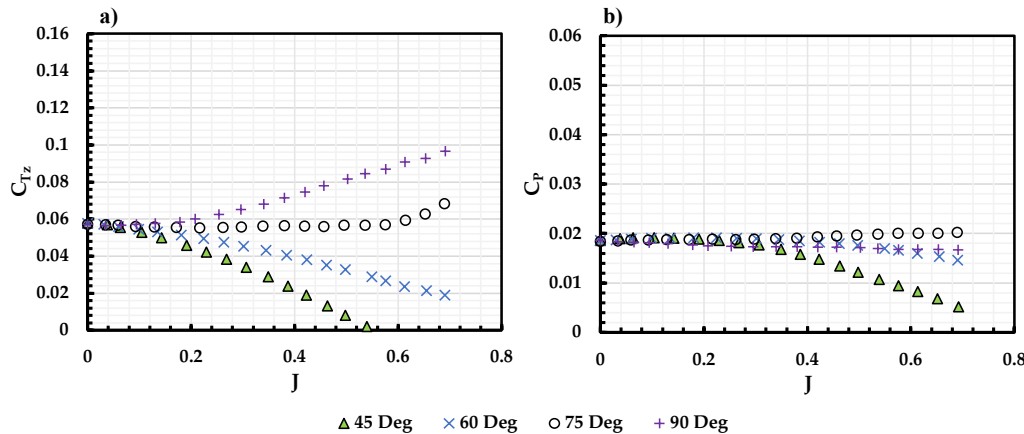

**Figure 9.** (**a**) $C_{Tz}$ and (**b**) $C_P$ vs $J$ for KDE 12.5 × 4.3 propeller under different $\theta$ at 100 RPS.

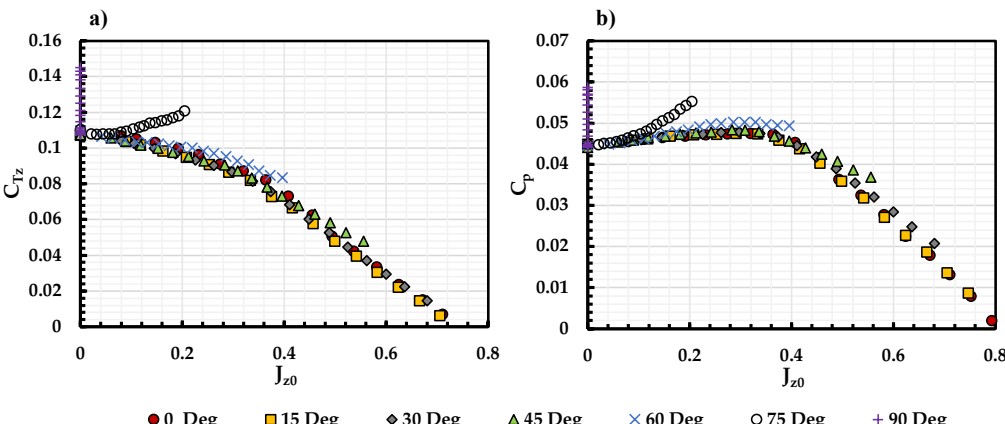

**Figure 10.** (**a**) $C_{Tz}$ and (**b**) $C_P$ vs $J_{z0}$ for APC 11 × 7 under different $\theta$ at 100 RPS.

The result is plotted in Figure 11 below for both 11 × 7 and 12.5 × 4.3 propellers. The efficiency for all $\theta$ cases collapses for the 11 × 7 propeller, except for $\theta = 90°$ where $J_z$ approaches to zero, which results in $\eta_z = 0$. While for the 12.5 × 4.3 propeller, the results collapse until $\eta_z$ reaches its maximum. Again, the KDE propeller is being optimized for edgewise flight which has a much lower performance at a lower incidence angle, where $C_{Tz}$ reduces significantly at small $\theta$ conditions, leading to a sudden drop of $\eta_z$ seen in Figure 11.

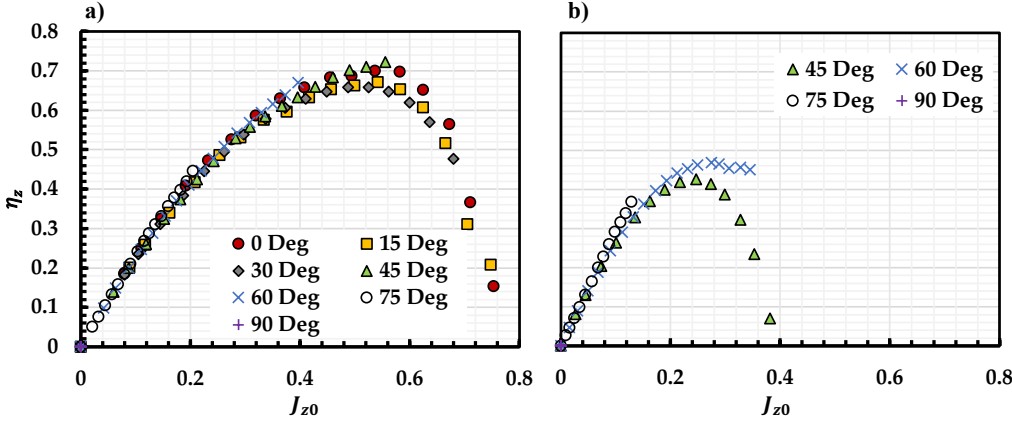

**Figure 11.** Propeller efficiency for (**a**) APC 11 × 7 and (**b**) KDE 12.5 × 4.3 calculated based on the normalized advance ratio $J_{z0}$.

Overall, the collapse of the $C_{Tz}$, $C_P$ and $\eta_z$ provides a relatively simple prediction model of propeller performance at $\theta < 60°$ using the data at $\theta = 0°$. Based on the results in Figure 11, it is evident that

$$C_{Tz}(J_{z0})_\theta = C_{Tz}(J)_{0°} \qquad \text{if } 0 < \theta < 60° \tag{4}$$

$$C_P(J_{z0})_\theta = C_P(J)_{0°} \qquad \text{if } 0 < \theta < 60° \tag{5}$$

The propeller pitching moment coefficient, $C_{Qx}$, and rolling moment coefficient, $C_{Qy}$, are shown as a function of $J$ for the APC $11 \times 7$ propeller under various $\theta$ in Figure 12. A positive $C_{Qx}$ represents a pitch-down moment and a positive $C_{Qy}$ represents a counter-clockwise moment. Different from the result in [22], $C_{Qx}$ decreases while $C_{Qy}$ increases with the increment in $J$ at $\theta > 0$, as a result of the asymmetric blade loading of a fixed pitch propeller. The advancing blade encounters a higher effective freestream velocity than the retrieving blade, while the blade pitch angle is held constant. This leads to a higher sectional lift and drag on the advancing blade than the retrieving blade. With the increment of either $J$ or $\theta$, this phenomenon will be amplified, as shown in Figure 12. Moreover, a noticeable reduction in slope of both $C_{Qx}$ and $C_{Qy}$ vs. $J$ curves is observed at $0° < \theta < 45°$ at $J \approx 0.4$. Recall from the APC $11 \times 7$ propeller results, $J \approx 0.4$ is the threshold where the freestream momentum begins to dominate the flowfield.

An alternative approach to normalizing $J$ at different $\theta$ can be borrowed from classical helicopter literature as shown in Equation (6).

$$J_{z90} = \frac{V sin(\theta)}{nD} \tag{6}$$

Similar to Equation (2), Equation (6) considers the freestream velocity in the propeller's normal direction and ignores the one in the propeller's axial direction. Using Equation (6), the moment curves are plotted with respect to $J_{z90}$ in Figure 13. $C_{Qx}$ collapses while $C_{Qy}$ collapses until deviating at different $J$ for different corresponding $\theta$. In this case, the $C_{Qx}$ and $C_{Qy}$ can be modeled by the freestream velocity in the propeller's normal direction. In other words, the propeller pitching moment is strongly related to the propeller inflow in its normal direction.

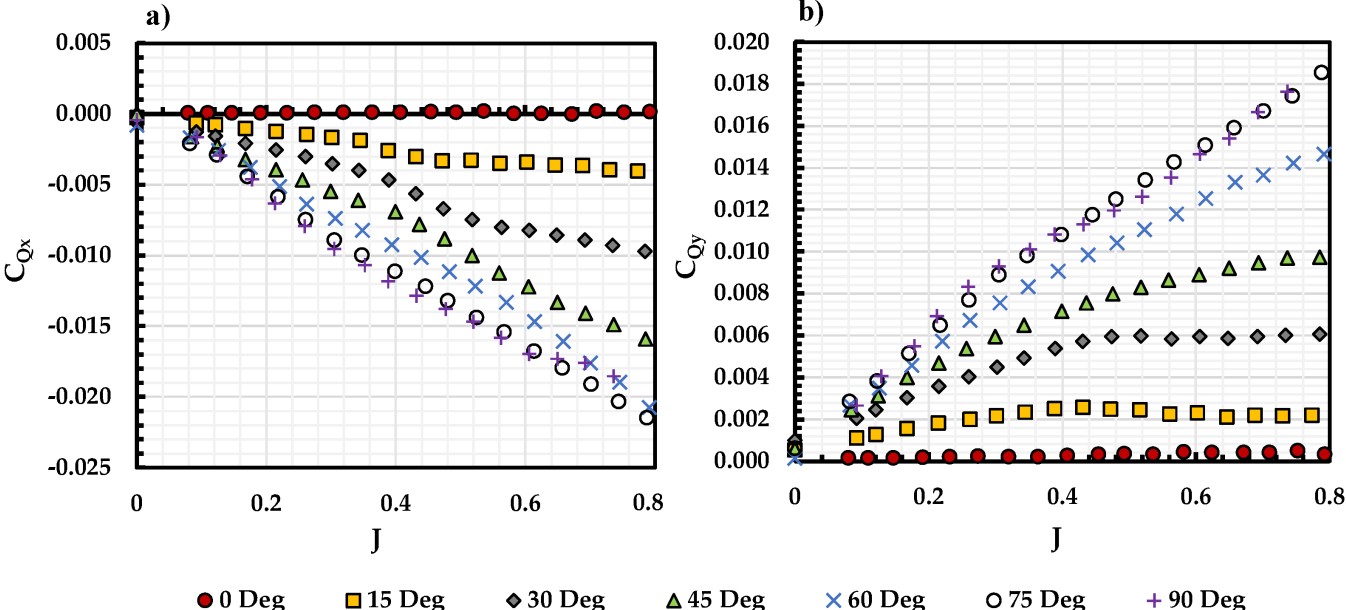

**Figure 12.** (**a**) $C_{Qx}$ and (**b**) $C_{Qy}$ for APC $11 \times 7$ propeller vs. $J$ at different incidence angles

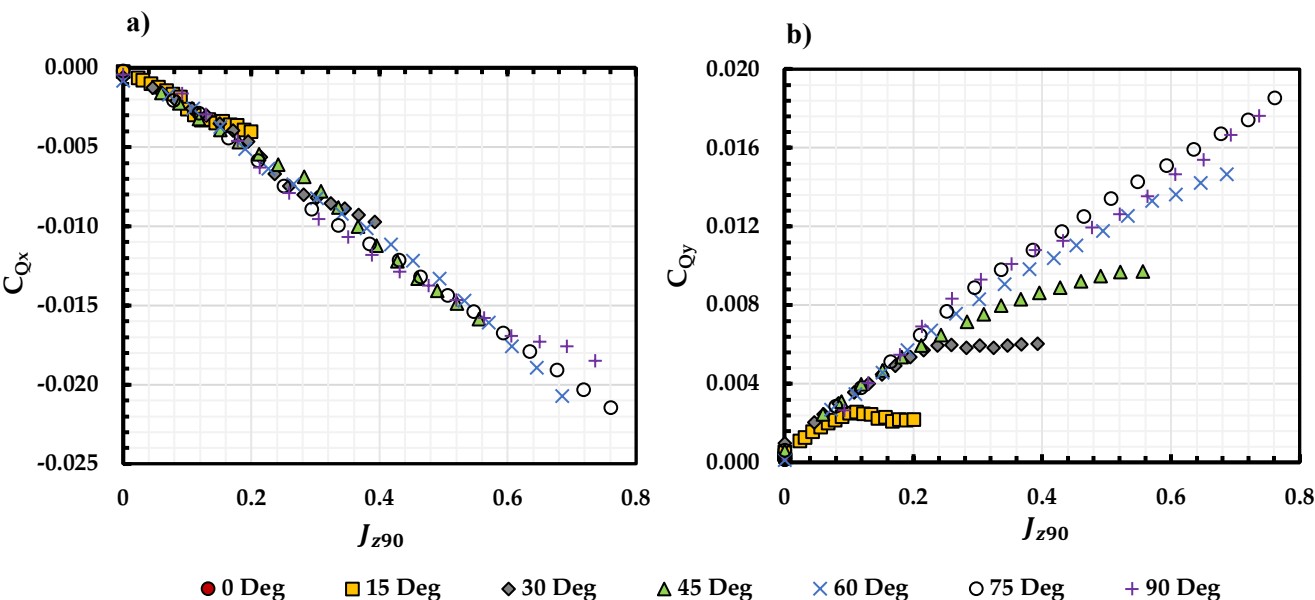

**Figure 13.** (**a**) $C_{Qx}$ and (**b**) $C_{Qy}$ for APC 11 × 7 propeller vs. $J_{z90}$ at different incidence angles.

*4.2. Propeller Performance under Time-Varying Sinusoidal Freestream*

The propeller thrust, power, and moment coefficients under unsteady freestream conditions will be discussed in this section. The unsteady propeller performance is also compared to a quasi-steady model that was developed from the steady freestream data discussed in the previous section. The time-varying velocity measured from the hot wire, along with the time-varying RPM results were used to determine the time-varying advance ratio. Then, the steady freestream results shown in Figures 8 and 9 was used to determine a time-varying $C_{Tz}$ and $C_P$.

4.2.1. KDE 12.5 × 4.3 Propeller

As mentioned earlier, KDE 12.5 × 4.3 propellers are commonly used on quad-copters in edgewise flight applications. The discussion in this section will focus on the unsteady response of $C_{Tz}$ and $C_{Qx}$ which mainly affects the overall stability of propeller-driven quadcopters.

Figure 14 shows the measured and predicted $C_{Tz}$, for the propeller at 83 RPS and in the wind tunnel fan rotation speed of $\Omega = 300$ RPM. The propeller was mounted at $\theta = 90°$. The measured freestream velocity is shown on the right Y-axis. Two propeller-reduced frequencies, $k_\omega = 0.012$ and $k_\omega = 0.294$ were selected for this analysis to show the propeller response at quasi-steady and unsteady conditions respectively. With the reduction of $U_\infty$, while maintaining the propeller rotational speed, the instantaneous $J$ for the propeller reduces, leading to an increment in $C_{Tz}$, as seen in Figure 9 for the steady freestream response. The steady-state model agrees extremely well with the measured $C_{Tz}$ at $k_\omega = 0.012$, indicating that the response is quasi-steady. However, at $k_\omega = 0.294$, a measurable phase lag can be observed between the experimental result and the steady-state model. A lower peak in the $C_{Tz}$ response is also measured when compared to the steady state model. Therefore, with an increase in reduced frequency, a measurable reduction in thrust coefficient and phase lag was observed indicating an unsteady response of the propeller.

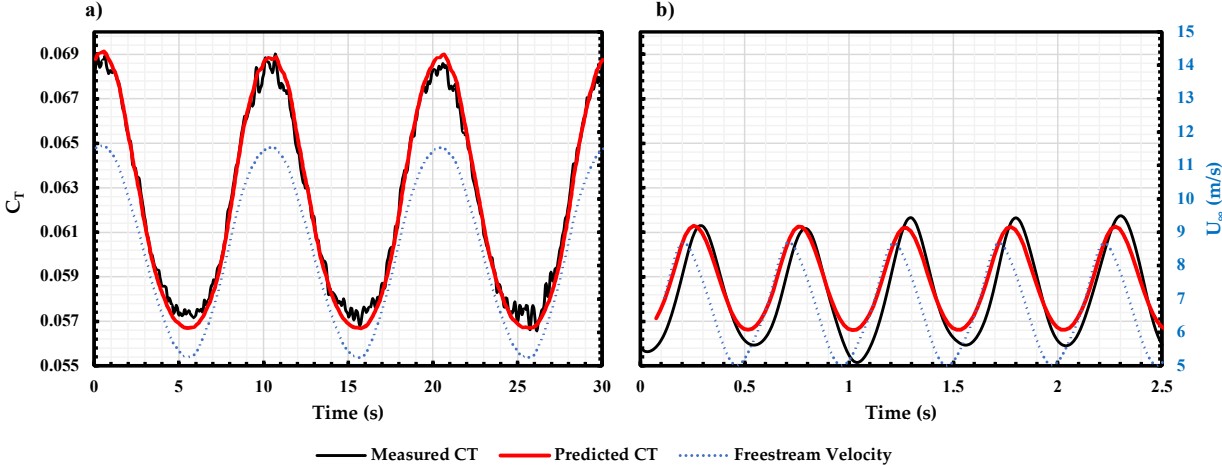

**Figure 14.** Measured and predicted $C_{Tz}$ at (**a**) $k_\omega = 0.012$ and (**b**) $k_\omega = 0.294$ for KDE 12.5 × 4.3 at 83 RPS, $\theta = 90°$ and $\Omega = 300$ RPM.

Figure 15 shows the $C_{Qx}$ response of the propeller under the same conditions as Figure 14. Again, the propeller response at $k_\omega = 0.012$ is quasi-steady and the steady state model overlaps on the measured $C_{Qx}$. While at $k_\omega = 0.294$, a phase lag and a change in the measured magnitude are observed due to the unsteady response of the propeller.

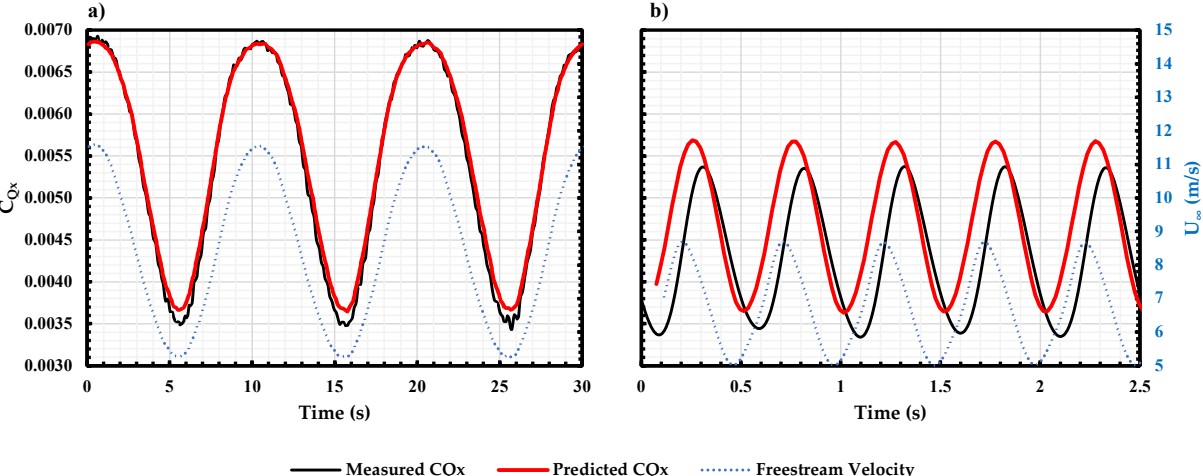

**Figure 15.** Measured and predicted $C_{Qx}$ at (**a**) $k_\omega = 0.012$ and (**b**) $k_\omega = 0.294$ for KDE 12.5 × 4.3 at 83 RPS, $\theta = 90°$ and $\Omega = 300$ RPM.

To better represent the phase lag of the propeller response, the instantaneous $C_{Tz}$ and $C_{Qx}$ v.s $J$ is shown in Figure 16 along with the steady-state performance (marked as the black dash line). At $k_\omega = 0.012$, the measured $C_{Tz}$ and $C_{Qx}$ overlaps on the steady-state performance curve, as seen in Figures 14 and 15. When increasing $k_\omega$ to 0.180, a hysteresis loop is seen for the measured $C_{Tz}$ and $C_{Qx}$ indicating a phase lag in the system. Further increase in $k_\omega$ to 0.294 resulted in a larger hysteresis loop. Moreover, a phase lag between the measured $C_{Tz}$ and measured $C_{Qx}$ is also observed due to the difference in the hysteresis loop. This can be seen better when plotting the measured $C_{Tz}$ vs. $C_{Qx}$ in Figure 17. A hysteresis loop occurred for the $C_{Tz}$ vs. $C_{Qx}$ plot at higher $k_\omega$ cases indicating the phase lag between $C_{Tz}$ and $C_{Qx}$.

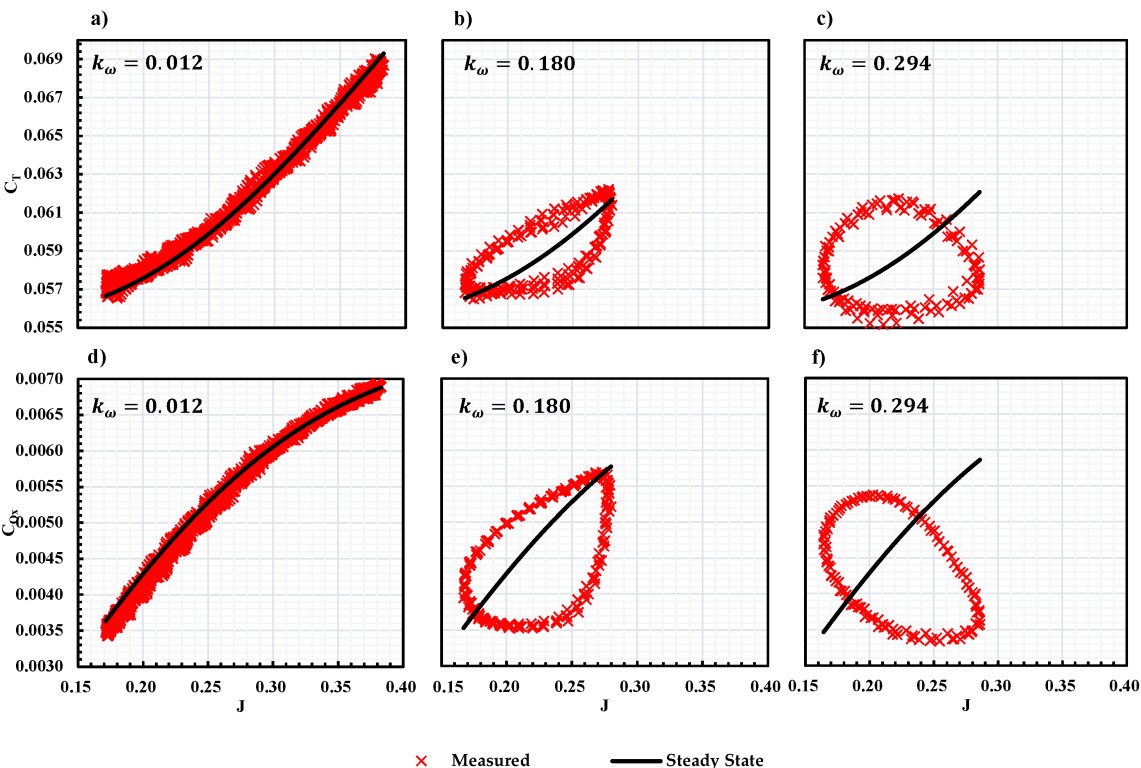

**Figure 16.** Measured and steady-state $C_{Tz}$ vs. $J$ (**a–c**) and $C_{Qx}$ vs. $J$ (**d–f**) for KDE 12.5 × 4.3 at 83 RPS, $\theta = 90°$ and $\Omega = 300$ RPM under $k_\omega$ of 0.012 (**a,d**), 0.180 (**b,e**) and 0.294 (**c,f**).

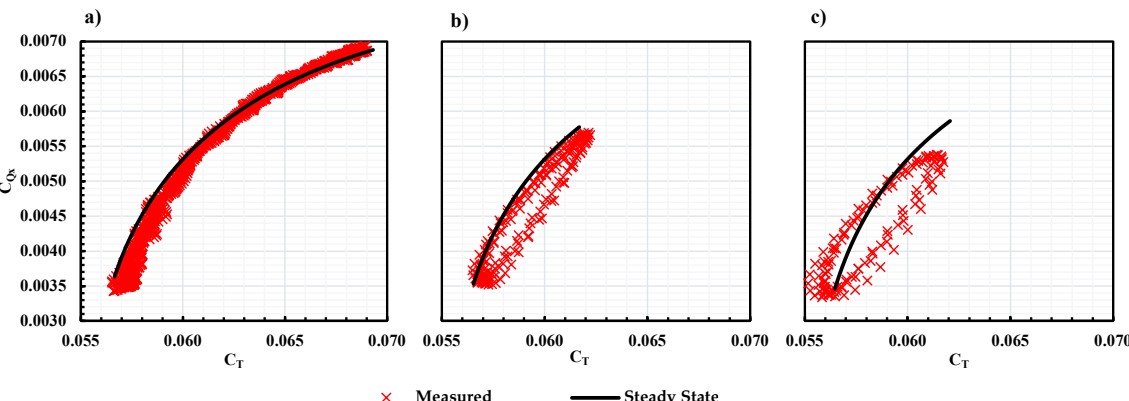

**Figure 17.** Measured and steady state $C_{Tz}$ vs. $C_{Qx}$ for KDE 12.5 × 4.3 at 83 RPS, $\theta = 90°$ and $\Omega = 300$ RPM under $k_\omega$ of (**a**) 0.012, (**b**) 0.180 and (**c**) 0.294.

Figure 18 summarizes the overall difference between the measured and steady-state $C_{Tz}$ and $C_{Qx}$. The mean value is marked as discrete data points while the upper and lower bound of the coefficient variation is marked as dashed lines. The bounds of the dashed lines are determined by the average difference between the peak and trough of the oscillation in the measured data.

The experimental results from two different tunnel speeds ($\Omega$) are shown in Figure 18 representing different reduced frequencies ($k_\omega$) range and advance ratio ($J$) for the propeller. At $\Omega = 200$ RPM, the mean velocity of the wind tunnel is lower than $\Omega = 300$ RPM as seen in Figure 7. Since the rotational speed of the propeller ($n$) is fixed, the resulting advance ratio at $\Omega = 200$ RPM is also lower when compared to $\Omega = 300$ RPM. As seen in Figure 7, a lower advance ratio would result in lower thrust and moment coefficients.

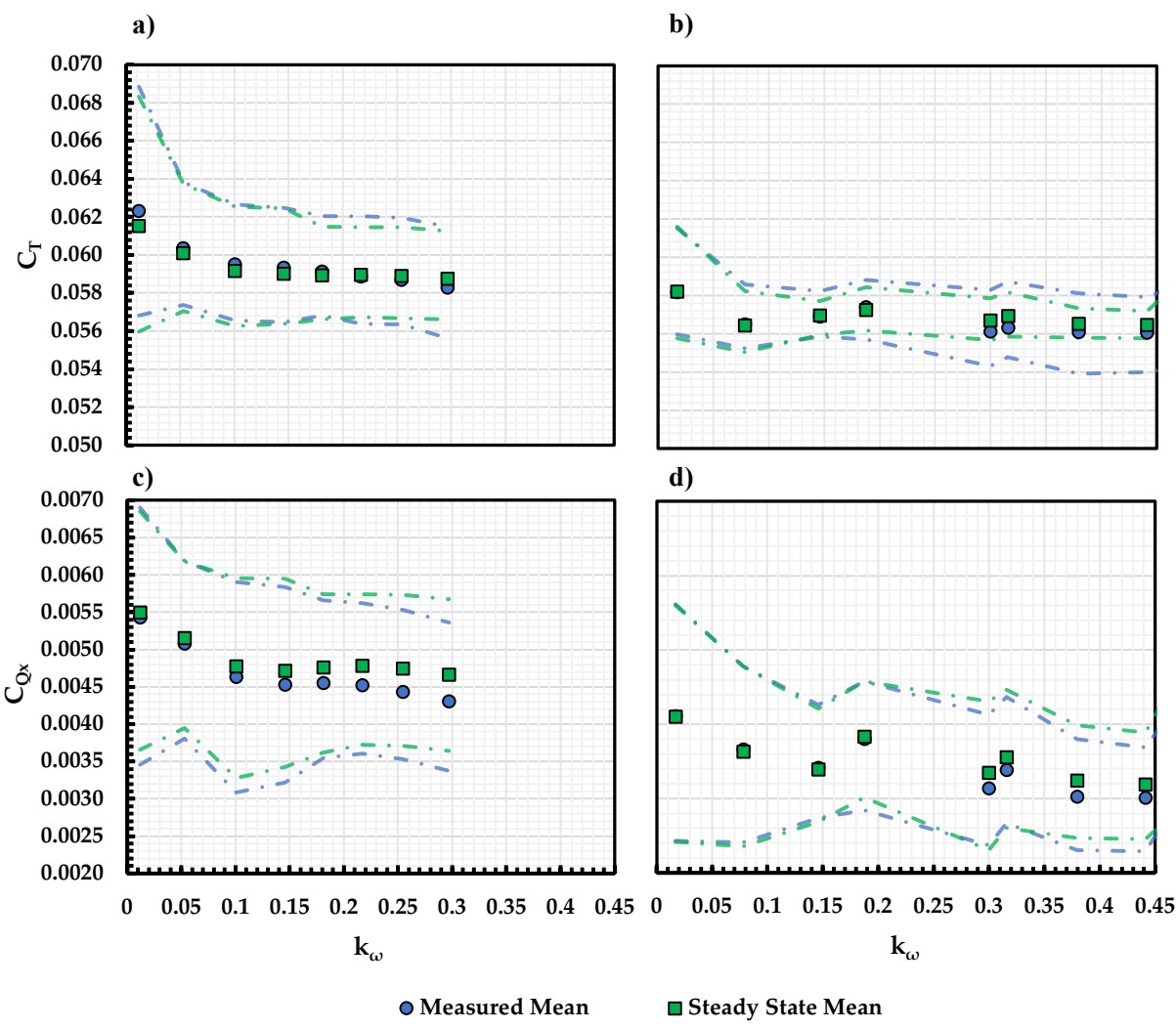

**Figure 18.** Measured and Steady State (**a**,**b**) $C_{Tz}$ and (**c**,**d**) $C_{Qx}$ mean and variation vs. $k_\omega$ for KDE 12.5 × 4.3 at 83 RPS, $\theta = 90°$ and (**a**,**c**) $\Omega$ = 300 RPM and (**b**,**d**) $\Omega$ = 200 RPM.

At $k_\omega \leq 0.2$, the mean value of the measured $C_{Tz}$ and $C_{Qx}$ matches with the steady state model relatively well as the datapoint overlaps. This is true for both $\Omega$ = 200 RPM and $\Omega$ = 300 RPM cases indicating that the unsteady propeller response is independent of the advance ratio. At $k_\omega > 0.2$, a small reduction in both measured mean values of $C_{Tz}$ and $C_{Qx}$ is observed when compared to the steady-state model. The results indicate that at higher reduced frequency gusts, the propellers will experience a loss in thrust along with a pitch-down moment and thus affecting the overall stability of the rotor.

4.2.2. APC 11 × 7 Propeller

Figure 19 shows the measured and steady state $C_{Tz}$ and $C_{Qx}$ for the APC 11 × 7 propeller at $\theta = 90°$ and $\Omega$ = 300 RPM. The overall trend follows that of KDE 12.5 × 4.3 shown in Figure 18. However, at $k_\omega > 0.2$, the measured $C_{Qx}$ and its variation is slightly greater than the steady state model indicating that the propeller tends to pitch up more. This is likely due to the difference in the propeller blade pitch as the APC 11 × 7 propeller has a higher blade pitch. In this case, the blade at the retreating side is more likely to stall in edgewise flight. Under unsteady conditions, this phenomenon is enhanced resulting in a higher pitch-up moment of the propeller disk.

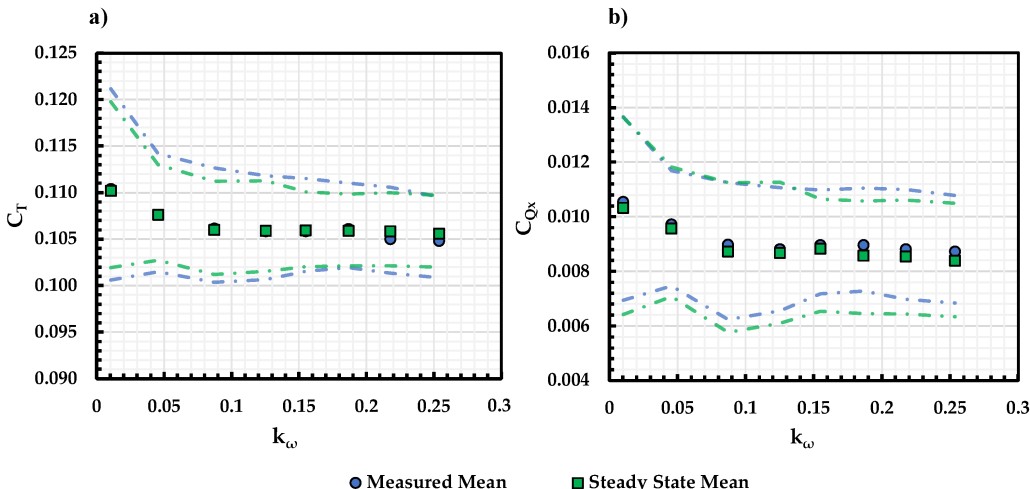

Figure 19. Measured and Steady State (a) $C_{Tz}$ and (b) $C_{Qx}$ mean and variation vs. $k_\omega$ for APC 11 × 7 at 83 RPS, $\theta = 90°$ and $\Omega$ = 300 RPM.

*4.3. Effect of the Propeller Incidence Angle*

4.3.1. KDE 12.5 × 4.3 Propeller

The results presented in the section above are for $\theta = 90°$ cases. However, in forward flight, the propeller is always at an incidence angle to the freestream. Therefore, the unsteady response of the propeller is investigated at $\theta = 75°$ and the results are shown in Figure 20 along with the results at $\theta = 90°$ under $\Omega$ = 200 RPM.

Similar to the $\theta = 90°$ cases, a reduction in both $C_{Tz}$ and $C_{Qx}$ is observed at $k_\omega > 0.2$ for the $\theta = 75°$ cases. It is worth noting that the variation in $C_{Tz}$ is much higher than the steady-state model. Recall from the steady state performance shown in Figure 9, the changes in $C_{Tz}$ with respect to $J$ is almost negligible at $\theta = 75°$ at $J < 0.55$. In this case, the steady-state model predicts a very small change in $C_{Tz}$ under streamwise gust when compared to the $\theta = 90°$ cases. On the hand, the response in $C_{Qx}$ also agrees with the result from $\theta = 90°$ cases in Figure 18, indicating that the trend is $\theta$ independent.

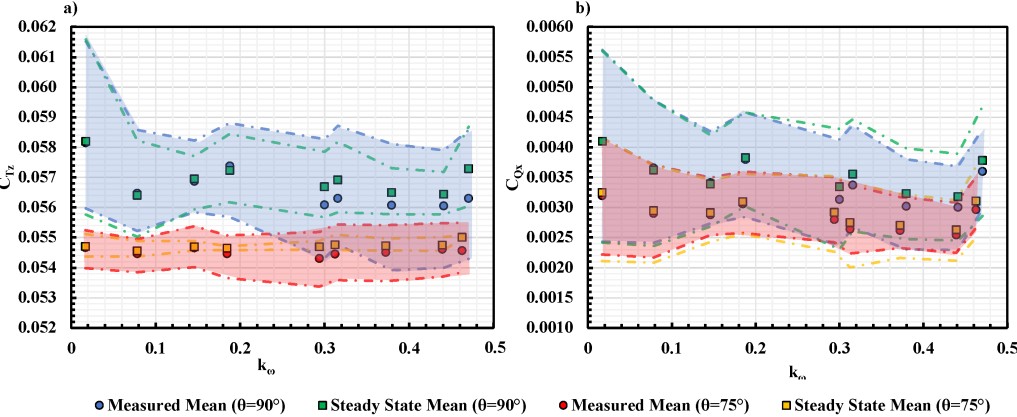

Figure 20. Measured and Steady State (a) $C_{Tz}$ and (b) $C_{Qx}$ mean and variation vs. $k_\omega$ for KDE 12.5 × 4.3 at 83 RPS, $\theta = 75°$ and $\Omega$ = 200 RPM.

4.3.2. APC 11 × 7 Propeller

The results for the APC 11 × 7 Propeller at $\theta = 75°$ are shown in Figure 21. Different from the result for the KDE 12.5 × 4.3 propeller in Figure 20, a higher measured $C_{Tz}$ and $C_{Qx}$ is observed for the APC 11 × 7 propeller at $k_\omega > 0.05$. As mentioned previously, the higher $\gamma/D$ propeller experiences blade stall at higher $\theta$. When encountering the streamwise gust, the flow reattachment occurs on the propeller blade resulting in a higher thrust generation which also leads to a higher pitching moment. This indicates that the

propeller performance under sinusoidal freestream might differ in forward flight based on the propeller geometry.

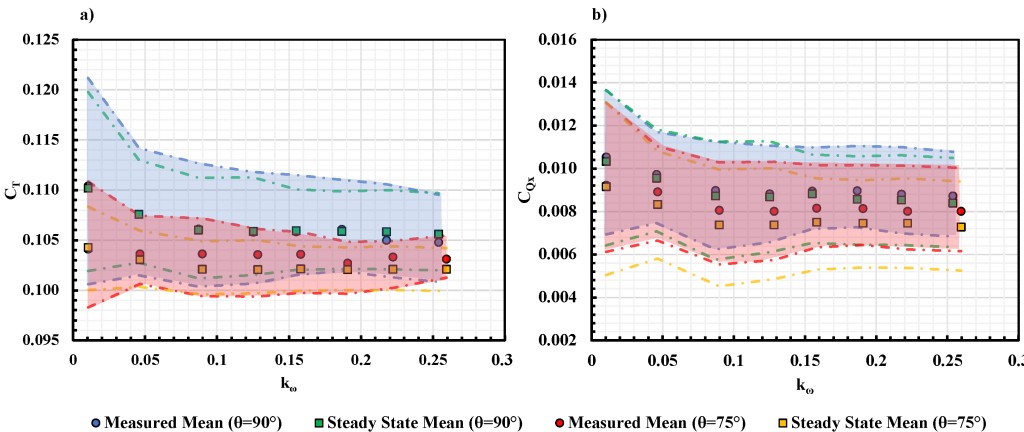

**Figure 21.** Measured and Steady State (**a**) $C_{Tz}$ and (**b**) $C_{Qx}$ mean and variation vs. $k_\omega$ for APC 11 × 7 at 83 RPS, $\theta = 75°$ and $\Omega = 300$ RPM.

### 4.4. Phase Response of Propellers

The overall $C_{Tz}$ and $C_{Qx}$ phase lag of the KDE 12.5 × 4.3 propeller under different experimental conditions discussed in this section is shown in Figure 22. The phase lag is measured by calculating the phase difference between the best fit sinusoidal wave of the measured data and the steady state model. The result from $\theta = 75°$ and $\Omega = 200$ RPM is excluded due to the nature of the propeller performance where the variation in $C_{Tz}$ is too small to identify the phase lag for this specific condition.

The phase lag for $C_{Qx}$ for all cases tested overlaps, despite the differences in $\theta$ and $J$. A significant phase lag is observed at $k_\omega \geq 0.2$ and an increase in phase of $-180°$ with the increment in $k_\omega$. However, the phase lag for $C_{Tz}$ differs between cases. At $\theta = 75°$, a phase lead is observed at $k_\omega < 0.3$, while such phase lead is not observed for the $\theta = 90°$. At $k_\omega > 0.2$, the magnitude of phase lag increases for both $\theta$ cases which are similar to the trend for $C_{Qx}$. Overall, the phase lag for the same $\theta$ cases overlaps with each other while for different $\theta$ cases, due to the aforementioned phase lead, the results do not overlap. It is also worth noting that the phase lag for $C_{Qx}$ is greater than $C_{Tz}$ for the same testing condition, which agrees with the results shown in Figures 16 and 17.

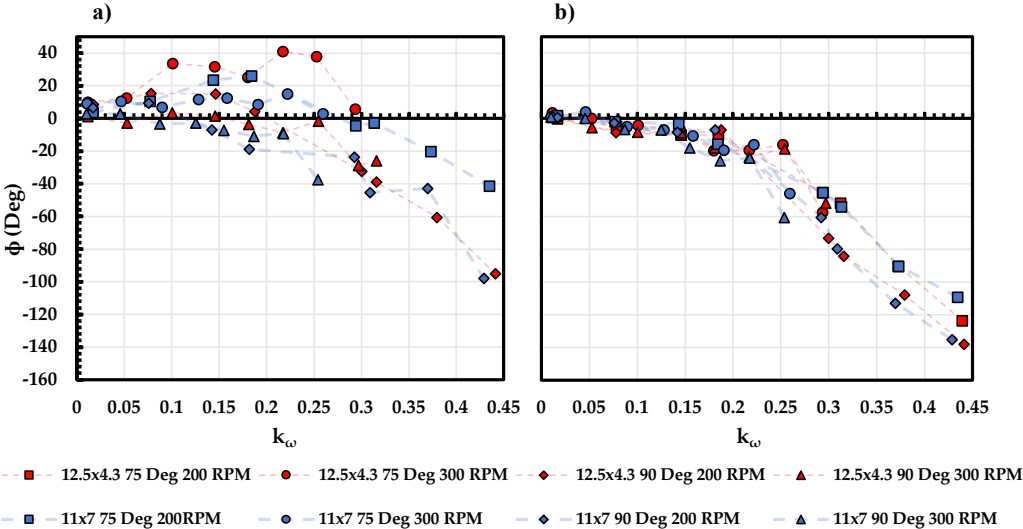

**Figure 22.** (**a**) $C_{Tz}$ and (**b**) $C_{Qx}$ phase lag for KDE 12.5 × 4.3 and APC 11 × 7 propeller under different $\theta$ and $\Omega$ at 83 RPS.

## 5. Conclusions

The frequency response of APC 11 × 7 and KDE 12.5 × 4.3 propellers in unsteady streamwise gust was experimentally quantified at two different incident angles. For the propeller steady-state response, we can conclude that:

1. An increment in propeller thrust, power, pitching moment, and rolling moment was found with the increment of incidence angle at the same advance ratio, which is consistent with the helicopter literature [9].
2. Using the normalized advance ratio, $J_{z_0}$ and $J_{z_{90}}$, the propeller performance under various incidence angles collapses, except for thrust and power coefficient in near edgewise flight conditions.

The propeller steady-state performance model was used to compare with the measured propeller thrust and pitching moment performance under unsteady freestream conditions, up to a reduced frequency of up to 0.45. For the propeller's unsteady response, we can conclude that:

1. A good fit between the steady-state model and measurement is found for both coefficients up to a reduced frequency of 0.2.
2. A reduction in both coefficients is found at a higher reduced frequency under 90° and 75° incidence angles for the lower $\gamma/D$ propeller. For the higher $\gamma/D$ propeller, an increment in both coefficients is observed at $\theta = 75°$.
3. A phase lag in the propeller response is also observed at a higher reduced frequency range. The phase lag for the pitching moment overlaps for all cases. While the phase lag for the propeller thrust depends on the incidence angle.
4. A reduction in the incidence angle leads to a phase lead in the thrust coefficient at a small reduce frequency range and a smaller phase lag at a higher reduced frequency range.

To sum up, it is reasonable to consider the propeller disk as a traditional 'flat plate' in unsteady aerodynamics studies, where classification to "highly unsteady" also occurs at a threshold reduced frequency over 0.2. In this case, the quasi-steady model can be used for the prediction of the propeller performance at a reduced frequency below 0.2, despite the differences in the propeller geometry and its flight condition. However, at a higher reduced frequency, the design of the propeller-driven UAV controller requires the consideration of both aforementioned propeller parameters. Moreover, the phase lag for the propeller response is significant at a higher reduced frequency, it would require a highly sophisticated control system that accounts for the phase lag. Future studies will focus on the response of the propeller in the multi-rotor configurations which will better represent the application of the propeller-driven UAVs.

**Author Contributions:** Conceptualization, S.G.; Methodology, J.C. and S.G.; Software, J.C.; Data curation, J.C.; Writing—original draft, J.C.; Writing—review & editing, S.G.; Supervision, S.G.; Project administration, S.G. All authors have read and agreed to the published version of the manuscript.

**Funding:** This research received no external funding.

**Data Availability Statement:** Data available on request due to restrictions.

**Acknowledgments:** The authors would like to thank Michael. V. OL for his advice throughout the study and Michael Mongin and Thomas Cook from the University of Dayton Low-Speed Wind Tunnel Research lab for their contribution in the early stages of the shuttering system characterization.

**Conflicts of Interest:** The authors declare no conflict of interest.

**Abbreviations**

The following abbreviations are used in this manuscript:

| | |
|---|---|
| $c$ | Propeller local chord length, ($m$) |
| $C_T$ | Propeller thrust coefficient; $C_T = T/(\rho n^2 D^4)$ |
| $C_P$ | Propeller Power coefficient; $C_P = P/(\rho n^3 D^5)$ |
| $C_Q$ | Propeller torque coefficient; $C_Q = Q/(\rho n^2 D^5)$ |
| $D$ | Propeller diameter, ($m$) |
| $J$ | Advance ratio; $J = V_\infty/(nD)$ |
| $J_z$ | Normalized advance ratio |
| $k_\omega$ | Normalized shuttering system frequency |
| $n$ | Propeller rotational speed per second |
| $P$ | Power, ($W$) |
| $Q$ | Torque, ($N.m$) |
| $T$ | Thrust, ($N$) |
| $\theta$ | Propeller incidence angle, ($deg$) |
| $\gamma$ | Propeller pitch, ($m$) |
| $\phi$ | Propeller local blade pitch angle, ($deg$) |
| $\omega$ | Shuttering system frequency, ($Hz$) |
| $\Omega$ | Wind tunnel fan rotational speed, ($RPM$) |
| $\Delta\lambda$ | Phase lag, ($rad$) |

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
