# Peer review of "Frequency Response of RC Propellers to Streamwise Gusts in Forward Flight"

_2674-032X, doi:10.3390/wind3020015_

Round 1

Reviewer 1 Report

1) The authors have written both 'RC' propeller (in Title) and 'R/C' propeller (in Keywords). It is suggested to adopt a uniform nomenclature. Better use the full form of 'RC'.

2) Important findings/results must be included in the ABSTRACT of the manuscript.

3) Add 'reduced frequency' as a keyword.

4) The literature/knowledge gaps should be mentioned clearly in the last paragraph of the Introduction section.

5) The measurement uncertainty and least count of the instruments involved should also be mentioned in Sec. 2.

6) The legibility of Fig. 4 (especially the schematic diagram of the experiment and the labeling associated with it) must be increased.

7) Mention the amount of deviation of the graphs in Fig. 5 to justify the terms 'well matched' and 'not well matched'.

8) In the Conclusions section, the key findings/understanding may be written pointwise (preferably in bulleted form) for clarity to the readers.

Author Response

Please see the file attached. 

Reviewer 2 Report

The authors report experimental studies of two different types of propellers under steady and sinusoidal time-varying free flow in an open jet wind tunnel. This article aims to compare the differences between these two situations. Considering that the real wind signal is also included in the effect of the gust, it is a good choice to use a special louver system. However, the quality of the article is not logical and organized enough, and the reviewers noticed some problems as follows.

Here are some detailed discussions:

1.      In section 2.1, mentioned two different sampled frequencies, respectively, 1000 Hz and 12500 Hz, but section 2.3 mentioned all the instrumentations are collected by LabView at 125 Hz, which made the reviewer confused, and the data sampled duration at 15 seconds, but Figure 14 a. shows a time-series until the 30s, it’s a bit strange.

2.      Most of the coefficients are not described and have clear definitions in the content, only put in the “Abbreviations”, which is very difficult to let the reviewer to understand the purpose of showing them in the results (i.e. : thrust coefficient, : torque coefficient et. al).

3.      The target input of the time-varying wind speeds is not shown, only mentioned the results match with the referred to sinusoidal model, but seems the louver frequency is not the same as the sinusoidal frequency, so it's difficult to understand the meaning of well matched. And here is used the ω to represent the shuttering system frequency, and the L325 is used again for the rotation speed of the wind tunnel’s fan. On the other hand, in usual ω is explain the angular frequency, but it does not describe the Hz, so it's easy to make the reader confused, expect this symbol, there are many similar mistakes in the definitions of the symbol which do not correspond on the context.

4.      In section 4.2.1, showed a series of results concerning the time-varying tests, Figure 14~17 can easier to see the results from different  are not with the same sampled time, so the results in Figure 18 are not convincing, because the time duration will affect the variation. The predicted curves seem from the steady-state results but here does show how it converts to a time series. Is it possible the phase lag is coming from the calculation error? For example, in Figure 14b and 15b, the first point of wind speeds are not started at a time equal to 0s.

Author Response

Please see the file attached. 

Reviewer 3 Report

This paper describes the R/C propeller performance under steady and sinusoidally time-varying using the University of Dayton Low-Speed Wind Tunnel (UD-LSWT) in the open-jet configuration. The propeller performance has been investigated in both steady and unsteady freestream conditions, and also been discussed along with the effect of inclination angle and propeller phase response. It shows that the propeller response of quasi-steady agrees with the prediction model below a reduced frequency of 0.2, and a reduction in mean thrust and pitching moment and significant phase lag at reduced frequencies higher than 0.2. The topic is of interest and the paper is generally well written. It could be considered for publication after some revisions. Please find below some points:

1)         It is supposed to have the full name before the abbreviation of UAVs in Line 25 since this abbreviation appears firstly.

2)         In Line 75-76, it is not very rigorous to say “To the extent of the authors’ knowledge……extremely sparse.” in an academic paper. There are some relevant literatures. Please go through the relevant publications and supplement these literatures to this section.

3)          It should replenish some literatures to prove the content described in Line 100 to 105.

4)         It would be better to restructure a bit of the Introduction section for showing the state-of-art more clearly.

5)         Which table is specified in Line 264?

6)         It is a bit confused of Figure 14, Figure 15. Why does the variation of freestream velocity appear along with the variations of thrust and torque coefficients in the same figure?

7)         Nomenclature should be used instead of Abbreviation in Line 440.

8)         The conclusion is a bit succinct and hasty, as the end of a technical report. It is better to extend it a bit with some further work and future studies.

Reviewer 4 Report

Please see the file attached. 
